# Clinical Characteristics of Molecularly Defined Renal Cell Carcinomas

**Xinfeng Hu, Congzhu Tan and Guodong Zhu \***

Department of Urology, The First Affiliated Hospital of Xi'an Jiaotong University, Xi'an 710061, China
* Correspondence: gdzhu@xjtufh.edu.cn; Tel.: +86-029-85323940

**Abstract:** Kidney tumors comprise a broad spectrum of different histopathological entities, with more than 0.4 million newly diagnosed cases each year, mostly in middle-aged and older men. Based on the description of the 2022 World Health Organization (WHO) classification of renal cell carcinoma (RCC), some new categories of tumor types have been added according to their specific molecular typing. However, studies on these types of RCC are still superficial, many types of these RCC currently lack accurate diagnostic standards in the clinic, and treatment protocols are largely consistent with the treatment guidelines for clear cell RCC (ccRCC), which might result in worse treatment outcomes for patients with these types of molecularly defined RCC. In this article, we conduct a narrative review of the literature published in the last 15 years on molecularly defined RCC. The purpose of this review is to summarize the clinical features and the current status of research on the detection and treatment of molecularly defined RCC.

**Keywords:** renal cell carcinoma; molecular; pathology; treatment; clinical

## 1. Introduction

Kidney cancer is the 14th most common cancer worldwide, and its incidence has continued to increase in recent years [1]. To date, more than 0.4 million new cases of kidney cancer are diagnosed each year [2,3]. Among them, more than 85% of patients present with renal cell carcinoma (RCC) [4]. Based on the traditional histopathological classification, RCC can be divided into three main categories: clear cell carcinoma (ccRCC, 75%), papillary renal cell carcinoma (PRCC, 15–20%), and chromophobe cell renal carcinoma (chRCC, 5%) [5]. Studies in recent years have found that RCC mostly occurs in older men [6] and most cases are localized tumors, with only 17% of RCC patients having distant metastases at the time of diagnosis, which are mainly found in lung, bone, liver, lymph nodes, and adrenal gland [1,4]. In 2020, a statistic by Padala SA showed that the 5-year survival rate for kidney cancer patients with metastatic disease was only 12% [7]. Currently, there are more and more treatment modalities for patients with metastatic RCC (mRCC), with targeted therapies and immune checkpoint inhibitor-based immunotherapy gradually proving to be effective in the treatment of patients with mRCC and the survival rate of those patients greatly improving recently [8,9].

Epigenetic alterations are considered to be a hallmark of cancer [10]. However, recent studies found that RCC has multiple molecular alterations, such as DNA methylation and micro-RNA alterations in ccRCC, which could greatly affect the biological progression of these tumors [11,12]. The 2022 World Health Organization (WHO) classification of pathological kidney tumors added new histopathological subtypes, including molecularly defined RCC [5,8]. It includes transcription factor binding to *IGHM* enhancer 3 (*TFE3*)-rearranged renal cell carcinomas, transcription factor *EB* (*TFEB*)-altered renal cell carcinomas, elongin C (*ELOC*)-mutated renal cell carcinoma, fumarate hydratase (*FH*)-deficient renal cell carcinoma, succinate dehydrogenase (*SDH*)-deficient renal cell carcinoma, anaplastic lymphoma kinase (*ALK*)-rearranged renal cell carcinomas, and *SWI/SNF*-related, matrix-associated,

actin-dependent regulator of chromatin subfamily B member 1 (*SMARCB1*)-deficient renal medullary carcinoma (see Table 1) [5]. These different molecularly defined histopathological subtypes of RCC are easily confused and may lead to suboptimal treatment outcomes as a result of misdiagnoses [12]. In this article, we summarize the pathological and clinical characteristics of each molecularly defined RCC subtypes and present their molecular features and the current treatment strategy status. We hope that this will be helpful for physicians to develop accurate diagnostic and therapeutic options for those RCC patients in clinical practice.

**Table 1.** Genes of molecularly defined renal cell carcinoma and associated clinical syndromes.

| Molecularly Defined Renal Cell Carcinoma Types | *TFE3*-Rearranged Renal Cell Carcinomas | *TFEB*-Altered Renal Cell Carcinomas | Elongin C (*ELOC*, Formerly *TCEB1*)-Mutated Renal Cell Carcinoma | Fumarate Hydratase-Deficient Renal Cell Carcinoma | Succinate Dehydrogenase-Deficient Renal Cell Carcinoma | *ALK*-Rearranged Renal Cell Carcinomas | *SMARCB1*-Deficient Renal Medullary Carcinoma |
|---|---|---|---|---|---|---|---|
| **Mutated genes** | Transcription factor binding to *IGHM* enhancer 3 (*TFE3*) | Transcription factor EB (*TFEB*) | Elongin C (*ELOC*) | Fumarate hydratase (*FH*) gene | Succinate dehydrogenase (*SDH*) | Anaplastic lymphoma kinase (*ALK*) | Subfamily B member 1 (*SMARCB1*) |
| **Location of genes** | Xp11.23 | 6p21 | 8q21.11 | 1q43 | SDHA: 5p15 SDHB: lp35-p36.1 SDHC: 1q21 SDHD: 11q23 | 2p23 | 22q11.2 |
| **Prevalence age** | Childhood | Childhood | Middle and old age | Adult | All ages | Childhood | Teenage |
| **Clinical Syndromes** | None | None | None | Hereditary leiomyomatosis and renal cell carcinoma (HLRCC) | *SDH*-deficient tumor syndrome | None | Rhabdoid tumor predisposition syndrome; familial schwannomatosis syndrome |
| **Chaperone genes** | *ASPL, PRCC, SFPQ, CLTC, PARP14, RBM10, NONO, MED15* | *MALAT1, CLTC, KHDRBS2, CADM2* | None | None | None | *VCL, TPM3, EML4, STRN, HOOK1* | None |
| **Mode of inheritance** | Dominant inheritance | Dominant inheritance | Dominant inheritance | Dominant inheritance | Dominant inheritance | Dominant inheritance | Dominant inheritance |
| **Morphological characteristics** | Transparent eosinophils; papillary architecture and psammoma bodies under the microscope | *TFEB*-translocated RCC: the biphasic growth pattern consisting of large and small tumor cells; smaller cells around the basement membrane-like structures; extensive hyalinization; papillary architecture; clear cell morphology. *TFEB*-amplified RCC: above pattern was less common | A clear cellular morphology under the microscope; thick fibromuscular bands; branching glandular vesicular; tubular structures | The papillary type or solid, tubulocystic, sieve-like type; abundant eosinophilic granulocytes, perinuclear halo | Cuboidal tumor cells, nested or tubular growth pattern. Characteristic morphology: the presence of vesicles or flocculent inclusions in the cytoplasm | *ALK*-rearranged RCC with *VCL* as a fusion gene: sickle-cell trait; eosinophilic granulocytic stroma; cytoplasmic lumen. Other *ALK*-rearranged RCC: similar to PRCC; consist of abundant intracellular and extracellular mucins; eosinophilic granuloplasm | At a high grade at the time of detection; infiltrative growth; sieve or reticular appearance |
| **Ancillary test (IHC, FISH)** | Positive: *PAX8* (100%); *TFE3* (95%); *CD10* (89%); achromatase (82%). Negative: *cytokeratin 7 (CK7); carbonic anhydrase 9 (CA9); GATA3* | Positive: *histone K; Melan-A*. *TFEB*-amplified RCC: diffusely or patchily positive when tested for *TFEB* levels | Positive: *CK7; ELOC; CA9; CD10; ELOC* in the nucleus. | Positive: *PAX8*; succinate dehydrogenase B abnormal succinate semicarbonate (2SC)S-(2-succino)-cysteine. Negative: *FH; CK7; TFE3* | Positive: *PAX8*; epithelial membrane antigen (*EMA*). Negative: *SDHB; CK7; CD117; histone K; TFE3;* HMB45. *SDHA*-deficient RCC showed negativity for *SDHA* | Positive: *PAX7; CK10; AMACR; CD3; cytokeratin; ALK.* Negative: *carbonic anhydrase IX; TFE45; histone enzyme K; Melan A;* HMB45 | Negative: *SMARCB1* |
| **Oncological behavior and prognosis** | May develop metastases within 20–30 years after diagnosis | *TFEB*-amplified RCC had higher tumor aggressiveness than *TFEB*-rearranged tumors. The 5-year survival rate for *TFEB*-amplified RCC was 48% | Has an aggressive oncological behavior | Have highly staged or distant metastases when diagnosed | Most cases are low grade and have a good prognosis with a low probability of metastasis | *ALK*-rearranged RCC with *VCL* as a fusion gene: no recurrence or distant metastasis. Other *ALK*-rearranged RCC: more aggressive clinical course | Often found at an advanced stage or with distant metastases; highly aggressive nature of the tumor. Average overall survival: 6–8 months |

## 2. TFE3-Rearranged Renal Cell Carcinomas

Transcription factor binding to *IGHM* enhancer 3 (*TFE3*) is an important regulator of the immune system and has now been shown to cooperate with transcription factor EB (*TFEB*) to control and regulate carbohydrate and lipid metabolism and mitochondrial homeostasis [13]. The *TFE3/TFEB* rearrangement renal cell carcinoma is characterized by translocations involving the *TFE3* and *TFEB* genes. They are both derived from the microphthalmia transcription (*MiT*) family of heterotopic RCC according to the 2016 version of the WHO classification. The *MiT* subfamily of transcription factors includes *TFE3*, *TFEB*, *TFEC*, and *MITF* [14]. *TFE3*- and *TFEB*-rearranged RCC accounts for 1–4% of the newly diagnosed adult patients [15]. Recent studies have shown that *TFE3/TFEB*-rearranged RCC can be frequently detected in children [16]. In adults RCC patients, *TFE3* ectopic fusions with chaperone genes are more commonly seen [17], and there are no significant prognostic gender differences [15] (Figure 1). This ectopic fusion with a chaperone gene and the decreased immunity in adults *TFE3*-rearranged RCC patients cause them to have a potentially more aggressive course compared to the pediatric patients [16]. Current studies suggest that previous exposure to cytotoxic chemotherapy might be a predisposing factor [18].

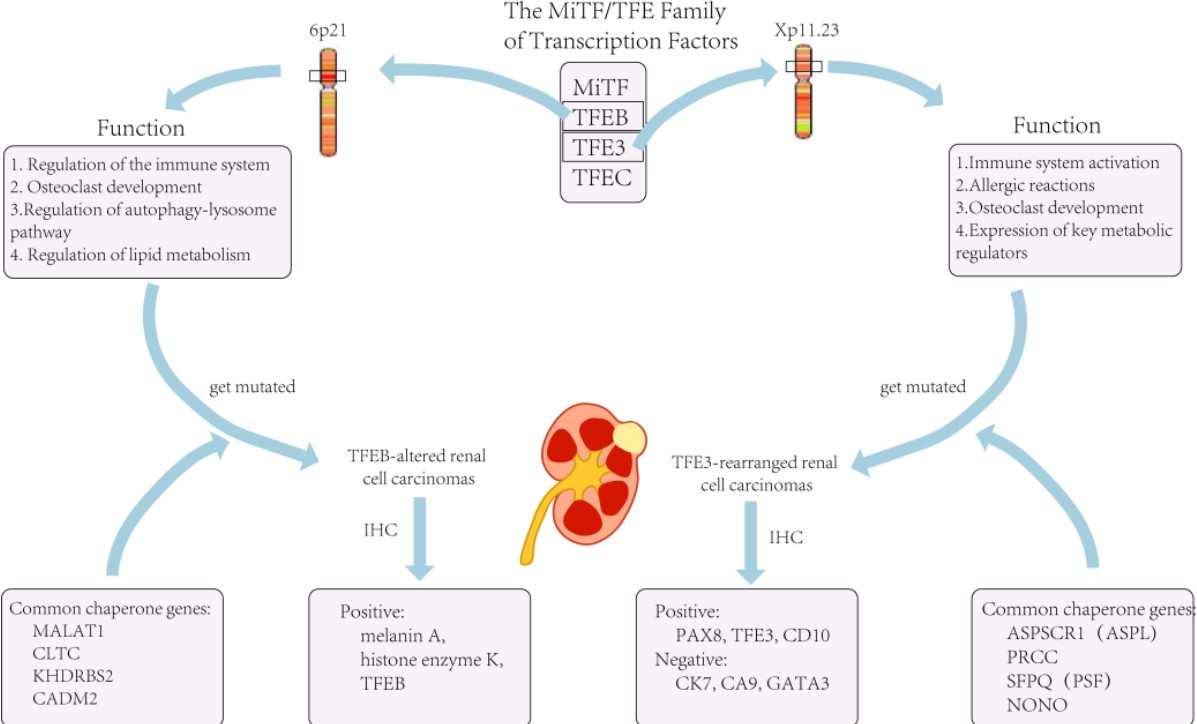

**Figure 1.** The role of *TFE3* in the organism and tumors caused by its mutation.

The list of chaperone genes has been growing and evolving, with more than a dozen having been reported [18]. The three most common translocations currently include a fusion of the *PRCC* and *TFE3* genes, a fusion of the *ASPL* (*ASPSCR1*) and *TFE3* genes, and a fusion of the *SFPQ* and *TFE3* genes [14]. In addition to this, there are also genes such as *CLTC*, *PARP14*, *RBM10*, *NONO*, and *MED15* that can be fused with ectopic *TFE3* [19]. However, current studies suggest that different chaperone genes may exhibit different oncological behaviors and tumor morphologies, and these features vary depending on the type of the involved chaperone genes [18]. For example, *TFE3* is more likely to exhibit lymph node metastasis when fused with *PRCC* than when fused with *ASPSCR3* [20].

In terms of histopathological morphology, the characteristics of *TFE3* fusion usually presents with transparent eosinophils, a papillary architecture, and psammoma bodies under the microscope [17,21]. However, due to chaperone genes, RCC with *TFE3* rear-

rangement may also resemble other types of RCC, including ccRCC, PRCC, and epithelioid vascular smooth muscle lipoma [14]. Therefore, attention should be paid and the impact of the genes that are fused with should be determined as much as possible both in the diagnosis and in the treatment of *TFE3*-rearranged renal cell carcinoma.

When facing *TFE3*-rearranged renal cell carcinoma, immunohistochemistry (IHC) is the most commonly used examination for diagnosis [13]. If IHC is not used at the time of diagnosis, a large proportion of TFE3-rearranged renal cell carcinomas are likely to be misdiagnosed as ccRCC [19]. For most other types of RCC, the positive IHC markers are *cytokeratin 7* (*CK7*), *carbonic anhydrase 9* (*CA9*), and *GATA3*. However, these are not expressed in *TFE3*-rearranged RCC and are usually positive for *histone K* [19]. In a recent review article, IHC data from nearly 400 cases of *TFE3*-rearranged RCC patients were analyzed, and the biomarkers with the highest probability of positivity were found to be *PAX8* (100%), *TFE3* (95%), *CD10* (89%), and *achromatase* (82%) [22]. However, *TFE3*-rearranged RCC did not always exhibit *TFE3* overexpression, and lower *TFE3* expression at the time of detection often resulted in false-positive or false-negative results; thus, this could limit the sensitivity and specificity of IHC for detecting *TFE3*-rearranged RCC [23]. In addition to this, the accuracy of IHC might be affected by the technique and be influenced by the formalin fixation time [18]. On the other hand, there has been no consensus or standardized guidelines regarding the judgment of *TFE3* staining results [18], and different pathologists might give completely opposite judgments if specimens show heterogeneous or focal staining. Lee HJ et al. used tissue specimens from 303 RCC patients for IHC testing and found that 23.2% of IHC-negative *TFE3* tumors were eventually diagnosed as *TFE3*-rearranged RCC [24]. Therefore, in clinical practice, a negative *TFE3* IHC result alone did not exclude the possibility of a *TFE3*-rearranged RCC case. Thus, in some cases, a combination of clinical presentation and other examination results might be needed.

The current literature suggests that the detection of *TFE3* gene rearrangements by fluorescence in situ hybridization (FISH) is more sensitive and advantageous in experimental manipulation than traditional IHC [23], and its results are more stable in formalin-fixed tissues [14]. Therefore, FISH is currently considered as the gold standard for the diagnosis of *TFE3*-rearranged RCC [23]. However, some chaperone genes, such as *NONO*, *RBM10*, and *GRIPAP1*, after fusing with *TFE3*, may not be detected by traditional FISH assays for significant *TFE3*-positive results. [18]. In addition, similar to IHC testing, the current standard definition of a positive FISH result varies widely among laboratories, from as low as 10% up to 30% [17]. These results suggest that, although the FISH test is currently the gold standard for the diagnosis of *TFE3*-rearranged RCC, in clinical practice, it should be carefully used together with other test results. For example, the previously mentioned IHC and FISH tests should be considered along with the option of gene probes or alternative molecular techniques [18]. In fact, FISH cannot provide information about fused genes, so in order to further confirm the diagnosis in clinical practice, RNA sequencing is often used to identify the gene involved in the translocation [17]. Recently, *TRIM63* determination by RNA in situ hybridization (RNA-ISH) was proposed as an alternative diagnostic tool for *TFE3*- and *TFEB*-rearranged RCC [25], but no strong evidence is available from in vitro studies.

Due to the rarity of *TFE3*-rearranged RCC and the fact that it has not been previously considered as a specific tumor subtype, there are no treatment recommendations for it to date [18]. Most previous treatment regimens are consistent with those for patients with ccRCC; however, due to recent developments in detection technology, its diagnosis has become more accurate, similarly to the detection of ccRCC. More importantly, drugs that normally treat ccRCC may not be effective against *TFE3*-rearranged RCC [16]. Additionally, Aldera AP et al. found that patients with *TFE3*-rearranged RCC may develop metastases within 20–30 years after diagnosis, so such patients may also need long-term clinical follow-up [26].

### 3. *TFEB*-Altered Renal Cell Carcinomas

As previously stated, *TFEB*-altered RCC has been included in the *MiTF*-translocated carcinoma family and *TFEB*-overexpressing renal tumors were initially identified in pediatric patients. Nowadays, with the availability of accurate examinations, more and more adult RCC patients are diagnosed with *TFEB*-altered RCC [27]. Nevertheless, the number of *TFEB*-altered RCC cases is still much lower than for *TFE3*-rearranged RCC [5]. There are two types of *TFEB*-altered RCC, including *TFEB*-rearranged RCC and *TFEB*-amplified RCC. The *TFEB* gene in *TFEB*-rearranged RCC is located on chromosome 6 and is most often translocated into chromosome 11, fusing with the *MALAT1* gene. Therefore, it was previously called t(6;11) RCC [19]. In the last few years, researchers have identified cases of RCC related to *TFEB* amplification, and after further testing and analysis, it was found that both genetic alteration patterns could co-exist in one case [5]. Due to the rarity of the disease, there are few studies on the distinction between different subtypes of *TFEB*-altered RCC, and current case studies show that the mean age of diagnosis for *TFEB*-amplified RCC is 62.5–64 years, while the mean age of diagnosis for *TFEB* translocated RCC is 32.8–34 years [28,29].

Similar to *TFE3*-rearranged RCC, *TFEB* can also be ectopically fused to chaperone genes [28] (Figure 1). Furthermore, for *TFEB*-amplified RCC, in addition to the possible elevated expression of *TFEB*, they are often accompanied by the amplification of other oncogenes, such as vascular endothelial growth factor A (*VEGFA*) and G1 S specific cyclin D3 (*CCND3*) [30]. It has been shown that these two genes are associated with aggressive oncological behavior [27], which would precisely explain the severe clinical symptoms and poor prognosis of patients with *TFEB*-amplified RCC.

Although *TFEB* genes are altered in *TFEB*-altered RCC, the characteristics of tumor growth vary considerably between different patterns of alteration. It has been widely reported that in *TFEB*-translocated RCC, the most commonly found morphology is a biphasic growth pattern consisting of large and small tumor cells [27], with smaller cells around the basement membrane-like structures. In addition to this, extensive hyalinization, a papillary architecture, and a clear cell morphology can be seen [31]. However, in *TFEB*-amplified RCC, this pattern is less common. Gupta S et al. investigated 37 patients with *TFEB*-altered RCC and found that nearly half of the patients had renal tubular structures and prominent cytoplasmic eosinophilia of tumor cells in their tumor specimens [27].

IHC and FISH are commonly used tests to detect *TFEB*-altered RCC; however, when assessing whether the *TFEB* gene is amplified or translocated, the markers used in the detection are quite similar. For *TFEB*-altered RCC, it has been shown that the staining results for both histone K and Melan-A are positive [31]. Similarly, Gupta S et al. and Wyvekens N et al. studied *TFEB*-amplified RCC and *TFEB*-translocated RCC, respectively, and they found that both types of tumors typically express melanin A and histone enzyme K. The difference was that tumor cells in *TFEB*-amplified RCC were usually diffusely or patchily positive when tested for *TFEB* levels [27]. However, there was also a subset of *TFEB*-amplified RCC that had lower *TFEB* expression levels than *TFEB*-translocated RCC [29]. Therefore, the type of *TFEB* gene alteration cannot be distinguished by a *TFEB*-specific assay alone. If a type of *TFEB* gene alteration is suspected, it should also be demonstrated using a FISH breakdown test or identified by RNA sequencing with a gene fusion examination [31]. In clinical practice, such detailed testing and diagnosis is not always necessary for all patients because of the very low incidence of the disease, the high cost of FISH, and the use of sequencing tests.

In addition to this, it has also been found that *TFEB*-amplified RCC exhibits a higher tumor aggressiveness than *TFEB*-rearranged tumors, and the 5-year survival rate for *TFEB*-amplified RCC is only 48% [32], while *TFEB*-translocated RCC progresses more slowly than *TFE3*-rearranged RCC. Therefore, in clinical practice, physicians should distinguish *TFEB*-amplified RCC from *TFEB*-translocated RCC. Since *TFEB*-altered RCC has often been previously diagnosed as ccRCC, its current treatment modality still differs little from the standard treatment for patients with ccRCC, which may also contribute to the poor

prognostic outcome for patients with *TFEB*-altered RCC. We hope that more appropriate targeted drugs and treatment strategies for *TFEB*-altered RCC will become available in the future.

## 4. *Elongin C* (*ELOC*, Formerly *TCEB1*)-Mutated Renal Cell Carcinoma

*Elongin C* (*ELOC*) is a transcription factor in the human body and the product of this gene expression is *ELOC*, which is part of the *von Hippel–Lindau* (*VHL*) protein complex and is responsible for the ubiquitination of hypoxia inducible factor 1 alpha subunit (*HIF1α*) and its subsequent degradation [33]. Previous studies have shown that *HIF* can activate the transcription of a large number of oncogenes, leading to tumorigenesis [34]. *ELOC*-mutated RCC was classified as ccRCC in previous WHO classifications [5], accounting for 0.5% to 5% of ccRCC [35]. However, in recent years, *ELOC*-mutated RCC has been found to present as wild-type *VHL*, exhibiting somatic mutations in the *ELOC* gene and deletion of the alternative allele (8q21) [36]. In addition to this, the microscopic morphology of *ELOC*-mutated RCC also differs in many ways from ccRCC [34]. Therefore, in the latest WHO classification for RCC, it was assigned to the molecularly defined tumors as a separate pathological type. *ELOC*-mutated RCC is a rare form of RCC [33] that usually develops in middle-aged and elderly male patients, most of whom are around 50 years of age [34].

Unlike the previous tumor types, *ELOC*-mutated RCC can be seen under the microscope with a clear cellular morphology [34]. It is usually similar to ccRCC [37], which has a transparent cellular appearance [34]. This is one of the reasons why it was assigned to the ccRCC category in the previous WHO classification. However, recent studies have shown that *ELOC*-mutated RCC also have thick fibromuscular bands and branching glandular vesicular or tubular structures similar to the morphology of ccRCC [37,38], and these manifestations can be distinguished from ccRCC. When tested using IHC, *ELOC*-mutated RCC can show the same aspects as ccRCC in that it is positive for both *CA9* and *CD10* [39]. However, in the study by Wang Y et al., IHC testing was performed in four patients with *ELOC*-mutated RCC and it was found that they all showed strong positive expression for *CA9* and three patients showed positive results for *CK7* and *CD10*. In addition, the authors observed *ELOC* positivity localized only in the nucleus of all four patients [34]. Despite the small number of cases selected, this result might also indicate that *ELOC* positivity in the nucleus was a characteristic manifestation of *ELOC*-mutated RCC. Similarly, Shah RB et al. conducted a study including 21 RCC patients with *ELOC* mutations and found that 16 of them had IHC staining results expressing diffuse positivity for *CK7* [39]. In summary, in addition to observing the characteristic structure of *ELOC*-mutated RCC under a microscope, the use of IHC to detect *CK7*, *ELOC*, *CA9*, and *CD10* could further help to confirm the diagnosis.

Previous studies have shown that *ELOC*-mutated RCC tends to be inert compared to ccRCC [36], but recently there have been some case studies demonstrating that certain cases could exhibit an aggressive oncological behavior. For example, DiNatale RG et al. investigated clinical data from five patients with *ELOC*-mutated RCC and found that four of them had advanced tumors (stage III-IV) and four had developed distant metastases [33]. This aggressiveness might be related to oncogene activation due to mutations in *ELOC*. Since *ELOC*-mutated RCC was previously widely considered to be one type of ccRCC, the current treatment is largely consistent with the treatment guidelines for ccRCC.

## 5. Fumarate Hydratase-Deficient Renal Cell Carcinoma

Fumarate hydratase (*FH*) is an indispensable enzyme in the tricarboxylic acid cycle that produces cellular energy in the form of ATP through oxidative phosphorylation (*OXPHOS*) in mitochondria [40]. Mutations in the gene where *FH* is located can lead to fumarate accumulation, which not only causes an imbalance in the energy supply but also impairs the function of histones and DNA demethylases, thus causing abnormal gene expression [41]. Singh NP et al. analyzed the TCGA database and found that alterations in the *FH* gene were associated with the immune function of *PRCC* [42], in addition to the accumulation

of metabolites, such as fumarate, which promote the expression of inflammatory factors and suppress the body's tumor immunity [43,44]. Fumarate hydratase (*FH*)-deficient RCC is a rare subtype of renal cancer that was considered a subtype of PRCC in the previous classification of RCC [40]. In the fifth edition of the WHO cancer classification, *FH*-deficient RCC has replaced hereditary leiomyomatosis and renal cell carcinoma (HLRCC) as a separate molecular subtype. It is characterized by germline mutations or somatic mutations in the *FH* gene, resulting in decreased expression of *FH* [45]. In addition to this, several studies have shown that methylation of genes, such as cyclin-dependent kinase inhibitor 2A (*CDKN2A*), O-6-methylguanine DNA methyltransferase (*MGMT*), adenomatous polyposis coli (*APC*), and tumor protein P53 (*TP53*), all of which are associated with tumorigenesis and progression, have been observed in *FH*-deficient RCC [45], which may explain the aggressiveness and poor prognostic outcome for patients observed in the clinic. HLRCC is an inherited syndrome caused by congenital mutations in *FH* gene and it is inherited in an autosomal dominant fashion. In clinical practice, HLRCC often presents as uterine tumors and smooth muscle tumors of the skin [5,40]. Indeed, it has long been shown that HLRCC increases the susceptibility to aggressive RCC [46]. However, there is no very precise treatment modality for patients with *FH*-deficient RCC.

Due to the rarity of *FH*-deficient RCC, the current knowledge of its disease characteristics and course is not very accurate. Yu YF et al. found that the mean age of onset was 36.7 years through a survey of 11 patients with *FH*-deficient RCC, which is lower than that of RCC patients without *FH* defects [47]. *FH*-deficient RCC could also exhibit many pathological structures, thus increasing its probability of being misdiagnosed [48]. Often, patients are much younger compared to other types of renal tumors when firstly diagnosed.

*FH*-deficient RCC can exhibit a variety of growth patterns and is, therefore, difficult to differentiate histologically [49]. The papillary type is the most common structure, and other common types include solid, tubulocystic, and sieve-like [47]. Microscopically, *FH*-deficient RCC also has characteristic histological manifestations, such as a papillary architecture with tubule cystic growth patterns, abundant eosinophilic granulocytes, and a perinuclear halo [40]. However, microscopic observation alone is not enough; more tests, such as IHC and imaging, are required to confirm the diagnosis [46]. In the clinical setting, genetic detection of mutations in *FH* is the gold standard for the diagnosis of *FH*-deficient RCC [50]. The imaging manifestations of *FH*-deficient RCC are very diverse, and it can present as a solid enhancing mass or as a mildly enhancing cystic mass, etc. These presentations cannot be distinguished from other types of RCC; therefore, diagnosis by imaging alone is incomplete [51]. Magnetic resonancespectroscopy (MRS) has also recently been proposed to be helpful in confirming the diagnosis of *FH*-deficient RCC. Wu G et al. used MRS in six patients with *FH*-deficient RCC and showed that the sensitivity, specificity, and accuracy were 69%, 100%, and 91%, respectively [52]. In IHC testing, the characteristic presentation of *FH*-deficient RCC is the lack of *FH* staining [45]; however, a recent study reported that there were isolated cases of *FH*-deficient RCC in which positive *FH* could still be detected [48]. Therefore, a positive result for *FH* does not completely exclude the possibility of *FH*-deficient RCC. In addition to detecting *FH*, studies in recent years suggested that some other biomarkers might play a key role in the detection of this disease. For example, *CK7* and *TFE3* usually show negative results, while *PAX8* and succinate dehydrogenase B abnormal succinate semicarbonate (2SC) S-(2-succino)-cysteine usually show positive results in the detection of patients with *FH*-deficient RCC [48,49,53].

Clinically, most *FH*-deficient RCC exhibit highly aggressive tumors, and patients are often found to have highly staged or distant metastases when they are diagnosed [54], with the most common sites of metastasis being the lymph nodes in the chest and abdomen, bone, and liver [55]. In addition, there is no clear standard treatment strategy for patients with *FH*-deficient RCC [45], and its highly aggressive course often makes treatment more difficult [46]. Most treatment stratigies for patients with *FH*-deficient RCC are quite similar to the treatment guidelines for patients withccRCC; however, due to the different pathogenesis and oncologic behavior, treatments that mimic ccRCC often result in an increased

chance of distant metastasis and death for patients with *FH*-deficient RCC [53]. In the past years, several new drugs have been explored for the treatment of this disease, such as sunitinib, pazopanib and immune checkpoint inhibitors (ICIs), including ipilimumab and nivolumab [46]. However, the efficacy of these drugs is not yet supported by clear positive evidence. In a recent study comparing treatment outcomes in 55 patients with *FH*-deficient RCC, the analysis found that the treatment with ICIs in combination with tyrosine kinase inhibitor (TKI) may have a better clinical outcome compared to monotherapy [56]. In addition to this, Gleeson JP et al. analyzed 26 patients with *FH*-deficient RCC to assess the efficacy of combined treatment with vascular endothelial growth factor (*VEGF*) and mammalian target of rapamycin (*mTOR*), and the study demonstrated that the objective response rate of this combination therapy was 44% [46]. In addition to this, recent reports demonstrated that bevacizumab in combination with erlotinib had entered phase II clinical trials and was currently showing positive results [47]. In the future, more targeted agents and more standard treatments will be available to help patients with *FH*-deficient RCC.

## 6. Succinate Dehydrogenase-Deficient Renal Cell Carcinoma

Succinate dehydrogenase (*SDH*), a complex that functions in mitochondria, is composed of several subunits (*SDHA*, *SDHB*, *SDHC*, and *SDHD*) [57]. It plays an important role in cellular respiration and energy metabolism, catalyzing the conversion of succinate to fumarate [58]. In tumorigenesis, *SDH* is considered as a class of cancer suppressor gene [57], and current studies demonstrate that when the *SDH* gene germline is altered, it often results in the development of paragangliomas, gastrointestinal mesenchymal tumors, and pituitary adenomas [59,60]. In addition, *SDH*-deficient RCC has also been shown to be associated with *SDH* germline mutations, and by far the most commonly found are mutations in *SDHB*, while *SDHC*, *SDHA*, and *SDHD* mutations are rare. *SDH*-deficient RCC is rare, accounting for an estimated 0.05% to 0.2% of all RCC cases [61].

*SDH*-deficient RCC can be seen in a wide variety of age groups and, in a survey by Gill AJ et al., they found that the age of diagnosis of *SDH*-deficient RCC can range from 14 to 76 years and is predominate in male patients [61]. Unlike the previously described RCC, most *SDH*-deficient RCC cases are low grade and have a good prognosis with a low probability of metastasis [58]. However, some *SDH*-deficient RCC cases with high-grade nuclei, sarcomatoid changes, or coagulative necrosis can have an aggressive oncological behavior with a poor prognosis [61]. Therefore, in facing RCC patients with the above pathological features, an aggressive molecular diagnosis should be clarified and early therapeutic measures should be taken to improve the quality of life and life expectancy of these patients.

For *SDH*-deficient RCC, its tumor cells are usually cuboidal, with nested or tubular growth pattern. However, its most characteristic morphology compared to other RCCs is the presence of vesicles or flocculent inclusions in the cytoplasm [58], which is often due to the enlargement of mitochondria as a result of an altered respiratory chain [59]. In terms of IHC, the negative result of *SDHB* staining is currently considered important for the definitive confirmation of the diagnosis [61]. However, recent studies have shown that decreased *SDH* expression is also observed in some non-*SDH* germline-deficient tumors [62], a condition that may be somewhat misleading in IHC, and, therefore, it may be inaccurate to solely rely on the decreased *SDH* expression to make the diagnosis. *SDH*-deficient RCC usually shows negativity for *CK7*, *CD117*, histone K, *TFE3*, and *HMB45*, but positivity for biomarkers such as *PAX8* and epithelial membrane antigen (*EMA*) [58,61,63]. Another recent study indicated that tumor cells in *SDHA*-deficient RCC showed negativity for both *SDHA* and *SDHB*, while RCC caused by defects in the *SDHB*, *SDHC*, or *SDHD* genes only showed negativity for *SDHB* [64]. This is also a possible way to diagnose *SDH*-deficient RCC accurately.

Clinically, most *SDH*-deficient RCC patients present as low-grade tumors; however, in some rare cases, distant metastases may be present [61]. In this regard, most *SDH*-deficient RCC can usually be easily cured by surgical resection [59], and for early-stage tumors,

even partial nephrectomy can be performed to preserve the kidney [58]. For patients with advanced-grade or with distant metastases, some studies have shown that targeted therapy with tyrosine kinase inhibitors, *VEGF*-targeted drugs, or *mTOR*-targeted drugs has shown positive therapeutic effects in patients with *SDH*-deficient RCC [65,66].

## 7. *ALK*-Rearranged Renal Cell Carcinomas

Anaplastic lymphoma kinase (*ALK*) is a membrane-associated tyrosine kinase that belongs to the insulin receptor family [67,68]. *ALK* functions to regulate cell proliferation and promote cell motility [69]. When *ALK* gene rearrangement occurs, it may lead to tumorigenesis. In 2011, two cases of *ALK*-rearranged RCC were first identified and diagnosed [70], and until now, it is still a very rare tumor [68], which accounts for 0.12–0.56% of all RCC cases [69]. Generally, a high expression of *ALK* can be observed in patients with *ALK*-rearranged RCC [71]. Similar to *TFE3*-rearraged RCC, it has many accompanying fusion genes. Various fusion genes have been identified in recent years, such as *VCL*, *TPM3*, *EML4*, *STRN*, and *HOOK1* [70], with renal tumors of *VCL* and *HOOK1* rearranged with *ALK* only described in pediatric patients [69].

Due to its rarity, there is no standard characteristic description of the clinical presentation of patients with *ALK*-rearranged RCC, which remains similar to PRCC and ccRCC in this sense [67]. *ALK*-rearranged RCC has many pathological manifestations, most of which display a shaped structure, in addition to solid and tubular patterns [70]. Among them, they can be roughly divided into two categories according to the morphology: One is *ALK*-rearranged RCC with *VCL* as a fusion gene, which occurs mostly in childhood and has a sickle-cell trait, eosinophilic granulocytic stroma, and cytoplasmic lumen [69,71]; the other category comprises other *ALK*-rearranged RCCs, most of which have a morphology similar to PRCC and also consist of abundant intracellular and extracellular mucins with eosinophilic granuloplasm [67,69]. In terms of IHC, the detection of *ALK* expressed in abundance in *ALK*-rearranged RCC using IHC has proven to be a valuable tool for the diagnosis of *ALK* [69]. In addition to this, several recent studies have found that the majority of *ALK*-rearranged RCC cases showed positive results for biomarkers such as *PAX7*, *CK10*, *AMACR*, *CD3*, and *cytokeratin*; negative results for biomarkers such as carbonic anhydrase IX, *TFE45*, histone enzyme K, Melan A, and *HMB45* [70,71]. These results can further help physicians to differentiate *ALK*-rearranged RCC from other types of RCC.

There is no standard treatment for patients with *ALK*-rearranged RCC; however, a recent study found that *ALK*-rearranged RCC with *VCL* as a fusion gene did not generally exhibit recurrence or distant metastasis [72], while *ALK*-rearranged RCC accompanied by other fusion genes showed a more aggressive clinical course [73]. As targeted agents continue to be developed, there is evidence that inhibitors of *ALK*, such as crizotinib and alectinib, can demonstrate efficacy in the treatment of nonsmall cell lung cancer and myofibroblastic tumors due to *ALK* rearrangements [74–76]. Although evidence for the treatment of *ALK*-rearranged RCC is still lacking, it is hoped that more clinical trials will be conducted in the future to demonstrate the efficacy of targeted agents for the treatment of patients with *ALK*-rearranged RCC.

## 8. *SMARCB1*-Deficient Renal Medullary Carcinoma

*SWI/SNF*-related, matrix-associated, actin-dependent regulator of chromatin subfamily B member 1 (*SMARCB1*) is a *SWI/SNF* protein complex that was considered to be a tumor suppressor in past studies and plays an important regulatory role in the organism [77]. In recent years, researchers have discovered that the *SMARCB1* gene is located on chromosome 22 and, when it is altered, *SMARCB1* expression is decreased or even absent [78], and a series of tumors are rapidly developed, such as malignant rhabdoid tumors of the central nervous system, renal medullary RCC, and epithelioid sarcoma [79]. In the 2022 edition of the WHO classification for RCC, this class of renal medullary carcinoma (RMC) with mutations in the *SMARCB1* gene is classified as a new molecular category called *SMARCB1*-

deficient RMC [5]. *SMARCB1*-deficient RMC is a rare cancer [80], which usually develops in patients with the sickle-cell trait (SCT) or sickle-cell disease (SCD).

*SMARCB1*-deficient RMC is an aggressive tumor that commonly affects males and is predominantly right sided [81]. It is often found at an advanced stage or with distant metastases, and recent studies have shown that *SMARCB1*-deficient RMC is also associated with the sickle-cell trait [82]. Specific symptoms are usually abdominal pain, hematuria, and weight loss [80], while distant metastases can be found in the renal lymph nodes, adrenal glands, lungs, and liver [83]. Due to the prevalence of *SMARCB1*-deficient RMC in children and adolescents and the aggressive nature of the tumor, early recognition and diagnosis are a priority for physicians.

In previous clinical practice, patients with *SMARCB1*-deficient RMC were often misdiagnosed as ccRCC [84]. With the advancement of detection technology in recent years, some characteristic manifestations of *SMARCB1*-deficient RMC have been gradually proposed. First, in addition to the previously mentioned clinical symptoms and prodromal nature during adolescence, *SMARCB1*-deficient RMC usually develops with SCT and SCD [83]. Secondly, the tumor is often already at a high grade at the time of detection, showing infiltrative growth and exhibiting a sieve or reticular appearance [85,86]. In addition to this, and most importantly, all *SMARCB1*-deficient RMC showed negative staining for *SMARCB1* when IHC for the detection of the *SMARCB1* protein was performed [83]. Therefore, when adolescent RCC patients with hematologic disorders such as SCT are identified in the clinic, physicians should perform IHC testing as early as possible to determine whether *SMARCB1*-deficient RMC is present.

Due to the rarity of the disease and the highly aggressive nature of the tumor, the current treatment options for *SMARCB1*-deficient RMC are not effective, with one study published in 2015 showing that the average overall survival of patients with *SMARCB1*-deficient RMC was only 6–8 months, with only one patient reaching 1 year [87]. Moreover, there is no standard treatment strategy for the disease. Due to the rapid progression of the disease, the predominant recommended treatment modality in the clinic is platinum-based chemotherapy [88]. In recent years, in addition to conventional treatments for kidney cancer, investigators have tried to explore the efficacy of various targeted agents for *SMARCB1*-deficient RMC. Examples include *VEGF* inhibitors, *mTOR* inhibitors (e.g., everolimus), etc. [83]; however, none of the patient outcomes have been very satisfactory. Immunosuppressive agents have been popular for oncology treatment, and Forrest SJ et al. tested 30 patients with *SMARCB1*-deficient RMC and found that 47% of them were positive for PD-L1 expression [89]. Furthermore, it has also been shown that, for *SMARCB1*-deficient RMC, differences in the tumor cell origin make it difficult for physicians to grasp the immune profile of the tumor [90]. Therefore, immunotherapy for *SMARCB1*-deficient RMC requires more in-depth studies in the future.

## 9. Conclusions

RCC is a common tumor that occurs mostly in men and most of them are low-grade tumors. However, in recent years, it has been discovered that RCC also has many specific molecular types, and the different molecular types may determine different clinical features and treatment outcomes. However, for many years, due to limited testing technology, many RCC patients were not diagnosed with a clear molecular type and most were managed according to the standard treatment protocol of ccRCC, resulting in poor outcomes and prognosis for many patients. This article presents the molecular types in the 2022 WHO classification of renal cancers, including the genetic alterations and clinical manifestations of each tumor type, followed by a summary of the current molecular testing results and current treatment status for each tumor type. Here, we suggest that urological clinicians should individualize the genetic level of testing when presented with RCC patients based on clinical manifestations and laboratory tests and should give targeted treatment after diagnosis. For certain congenital genetic defective RCCs, attention should also be paid to the effect of the genetic defect at other sites. However, because physicians did not

previously pay much attention to the molecular types of kidney cancer, and because of the rarity of the onset of certain RCC types, the existing clinical studies are inevitably limited in terms of sample size, observation angle, and treatment bias, and there are still many inconsistent conclusions on the characteristic manifestations of molecular detection and clinical treatment criteria. In recent years, research on molecular detection technologies and targeted drugs or immune checkpoint inhibitors has progressed very rapidly, physicians' knowledge of the disease has become more and more mature, and significant progress has been made in the diagnosis and treatment of RCC. In the future, we hope that there will be more tests and detection standards for RCC in molecular science and effective drugs to help RCC patients have a better prognosis and higher quality of life.

**Author Contributions:** Conceptualization, G.Z.; original draft writing, X.H.; figures preparation and table making, C.T.; manuscript review and editing, G.Z.; funding acquisition, G.Z. All authors have read and agreed to the published version of the manuscript.

**Funding:** The Fundamental Research Funds for the Central Universities of China (No. xjj2018zyts34) and the Research Funds on Social Development from the Department of Science and Technology of Shaanxi Province of China (No. 2020SF-119) to Guodong Zhu are acknowledged.

**Institutional Review Board Statement:** Not applicable.

**Informed Consent Statement:** Not applicable.

**Data Availability Statement:** Not applicable.

**Conflicts of Interest:** The authors declare no conflict of interest.

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
