# Peer review of "Clinical Characteristics of Molecularly Defined Renal Cell Carcinomas"

_cimb, doi:10.3390/cimb45060303_

Round 1

Reviewer 1 Report

The review can be a helpful and informative article after some modifications. Since the aim of the review is to summarize the pathological and clinical characteristics, please elaborate more information about these two aspects for each subtype.

1.       Abstract: “However, researches on these types of RCC are still incomplete,…”

Comments: What does it mean with incomplete researches?

2.       Table 1: Please correct the chaperonr to chaperone

3.       Please include columns with a summary of information below in table 1:

A.      Morphological characteristics

B.       Ancillary test (IHC, FISH)

C.       The presence of metastasis

D.      Overall survival and disease-free survival

4.       From what I understand, each subchapter contains information as follows:

A.                  The subtype; the genetic alterations and the gene functions background

B.                   WHO classification

C.                   Age of patients

D.                  Clinical signs and symptoms (including syndromes)

E.                   Morphological features

F.                   Ancillary test (IHC, FISH, etc) to confirm the diagnosis of subtype

G.                  The presence of metastasis

H.                  Overall survival or DFS of the patients

I.                     Targeted therapy of the patient

Please use a consistent order and format style among different subchapters. Please ensure all the points are discussed in each subchapter in order to fairly compare RCC subtypes.

 Minor editing of English language required especially for the typos.

Author Response

Point 1:  Abstract: “However, researches on these types of RCC are still incomplete,…” Comments: What does it mean with incomplete researches?

Response 1: Thank you very much for your comment. After careful literature retrieval, we believe that the current research on those molecularly defined RCC is not comprehensive enough, and there is still a lot of uncertainty from diagnosis to treatment in clinical practice. We have corrected this statement and added detailed views in the abstract part of the revised manuscript.

Point 2:  Table 1: Please correct the chaperonr to chaperone

Response 2: Thank you for reminding us this error, and we have corrected this mistake in table 1 of the revised manuscript.

Point 3:  Please include columns with a summary of information below in table 1:

  1. Morphological characteristics
  2. Ancillary test (IHC, FISH)
  3. The presence of metastasis
  4. Overall survival and disease-free survival

Response 3: Thank you very much for your comment, and we have added items A and B in revised table 1. After careful literature retrieval, we found that there is no available literature or data on items C and D in those molecularly defined RCC. Therefore, we combined the C and D items into a new item: “Oncological behavior and prognosis.” in the revised table 1.

Point 4:  From what I understand, each subchapter contains information as follows:

  1. The subtype; the genetic alterations and the gene functions background
  2. WHO classification
  3. Age of patients
  4. Clinical signs and symptoms (including syndromes)
  5. Morphological features
  6. Ancillary test (IHC, FISH, etc) to confirm the diagnosis of subtype
  7. The presence of metastasis
  8. Overall survival or DFS of the patients
  9. Targeted therapy of the patient

Please use a consistent order and format style among different subchapters. Please ensure all the points are discussed in each subchapter in order to fairly compare RCC subtypes.

Response 4: Thank you very much for your comment, we have carefully reviewed the manuscript and rearranged them in the order as described above. Please see part 3, part 4 and part 5 of the revised manuscript.

Reviewer 2 Report

There are several areas in the manuscript that need some editing to make the language clearer.  Here are few of these errors - You state SMARCB1-overexpressing RMC in the third sentence of the renal medullary carcinoma section, I am assuming this is an error and should state SMARCB1 loss. The first sentence on page 10 should read  - SMARCB1-deficient RMC usually develops in patients with sickle cell trait or sickle cell disease.  In the conclusions section, I would use RCC for all abbreviations not a mixture of RCC and RC. 

A table with the IHC markers and/or FISH assay used for diagnosis for each molecularly defined renal carcinoma subtype should be added. This table should include the positive and negative markers for each molecular subtype. 

In addition, several publications need to be added.    

To the introduction – these publications should be added

Padala SA, Barsouk A, Thandra KC, Saginala K, Mohammed A, et al. Epidemiology of Renal Cell Carcinoma. World J Oncol. 2020;11(3):79-87.

Bray F, Ferlay J, Soerjomataram I, Siegel RL, Torre LA, Jemal A. Global cancer statistics 2018: GLOBOCAN estimates of incidence and mortality worldwide for 36 cancers in 185 countries. CA Cancer J Clin. 2018;68(6):394-424.

In the renal medullary carcinoma section – this publication

Msaouel P, Malouf GG, Su X, Yao H, Tripathi DN, et al. Comprehensive Molecular Characterization Identifies Distinct Genomic and Immune Hallmarks of Renal Medullary Carcinoma. Cancer Cell. 2020;37(5):720-734 e713.

Some spelling mistakes were found throughout the manuscript and other language choice errors.  

Author Response

Point 1: There are several areas in the manuscript that need some editing to make the language clearer.  Here are few of these errors -You state SMARCB1-overexpressing RMC in the third sentence of the renal medullary carcinoma section; I am assuming this is an error and should state SMARCB1 loss.

Response 1: Thank you for reminding us this error. After careful literature retrieval, we have replaced "SMARCB1-overexpressing RMC" with "RMC with mutations in SMARCB1 gene" in part 8 of the revised manuscript.

Point 2:  The first sentence on page 10 should read- SMARCB1-deficient RMC usually develops in patients with sickle cell trait or sickle cell disease.

Response 2: Thank you for reminding us this error, and we have revised this sentence in the revised manuscript as suggested.

Point 3: In the conclusions section, I would use RCC for all abbreviations not a mixture of RCC and RC. 

Response 3: Thank you very much for your comment. We have carefully reviewed the manuscript and modified all RC to RCC in the conclusions section.

Point 4: A table with the IHC markers and/or FISH assay used for diagnosis for each molecularly defined renal carcinoma subtype should be added. This table should include the positive and negative markers for each molecular subtype. 

Response 4: Thank you very much for your comment. After careful literature retrieval, we have added these items in revised table 1.

Point 5: In addition, several publications need to be added.    To the introduction – these publications should be added Padala SA, Barsouk A, Thandra KC, Saginala K, Mohammed A, et al. Epidemiology of Renal Cell Carcinoma. World J Oncol. 2020;11(3):79-87.

Bray F, Ferlay J, Soerjomataram I, Siegel RL, Torre LA, Jemal A. Global cancer statistics 2018: GLOBOCAN estimates of incidence and mortality worldwide for 36 cancers in 185 countries. CA Cancer J Clin. 2018;68(6):394-424.

In the renal medullary carcinoma section – this publication: Msaouel P, Malouf GG, Su X, Yao H, Tripathi DN, et al. Comprehensive Molecular Characterization Identifies Distinct Genomic and Immune Hallmarks of Renal Medullary Carcinoma. Cancer Cell. 2020;37(5):720-734 e713.

Response 5: Thank you very much for your comment. After careful literature retrieval and reading, we have cited these articles which are mentioned above as the important references in introduction part and the renal medullary carcinoma section of the revised manuscript.